# Hybrid Coronary Percutaneous Treatment with Metallic Stents and Everolimus-Eluting Bioresorbable Vascular Scaffolds: 2-Years Results from the GABI-R Registry

**DOI:** 10.3390/jcm8060767

**Published:** 2019-05-30

**Authors:** Tommaso Gori, Stephan Achenbach, Thomas Riemer, Julinda Mehilli, Holger M. Nef, Christoph Naber, Gert Richardt, Jochen Wöhrle, Ralf Zahn, Till Neumann, Johannes Kastner, Axel Schmermund, Christian Hamm, Thomas Münzel

**Affiliations:** 1Zentrum für Kardiologie, University Medical Center, Johannes Gutenberg University Mainz, 55131 Mainz, Germany; tmuenzel@uni-mainz.de; 2German Centre for Cardiovascular Research, partner site Rhine Main, 55131 Mainz, Germany; 3Department of Cardiology, Friedrich-Alexander University Erlangen-Nürnberg, 91054 Erlangen, Germany; achenbach@uni-erlangen.de; 4IHF GmbH-Institut für Herzinfarktforschung, 67063 Ludwigshafen, Germany; riemer@ihf.de; 5Department of Cardiology, Munich University Clinic, LMU, 80539 Munich, Germany; mehilli@lmu.de; 6German Centre for Cardiovascular Research, partner site Munich Heart Alliance, 80539 Munich, Germany; 7Department of Cardiology, University of Giessen, Medizinische Klinik I, 35392 Giessen, Germany; h.nef@me.com (H.M.N.); hamm@neuheim.de (C.H.); 8Klinik für Kardiologie und Angiologie, Elisabeth-Krankenhaus, 45138 Essen, Germany; naber@contilia.de; 9Herzzentrum, Segeberger Kliniken GmbH, 23795 Bad Segeberg, Germany; richrdt@segeberg.de; 10Department of Internal Medicine II, University of Ulm, 89081 Ulm, Germany; woehrle@ulm.de; 11Abteilung für Kardiologie, Herzzentrum Ludwigshafen, 67063 Ludwigshafen, Germany; zahn@klinikum.de; 12Department of Cardiology, University of Essen, 45138 Essen, Germany; neumann@uniwien.at; 13Department of Cardiology, University of Vienna Medical School, 1090 Wien, Austria; kastner@viennw.at; 14Bethanien Hospital, 60389 Frankfurt, Germany; schmermund@ccb.de; 15Department of Cardiology, Kerckhoff Heart and Thorax Center, 61231 Bad Nauheim, Germany

**Keywords:** coronary artery disease, drug eluting stents, stent bioresorbable

## Abstract

The limitations of the first-generation everolimus-eluting coronary bioresorbable vascular scaffolds (BVS) have been demonstrated in several randomized controlled trials. Little data are available regarding the outcomes of patients receiving hybrid stenting with both BVS and drug-eluting stents (DES). Of 3144 patients prospectively enrolled in the GABI-Registry, 435 (age 62 ± 10, 19% females, 970 lesions) received at least one BVS and one metal stent (hybrid group). These patients were compared with the remaining 2709 (3308 lesions) who received BVS-only. Patients who had received hybrid stenting had more frequently a history of cardiovascular disease and revascularization (*p* < 0.05), had less frequently single-vessel disease (*p* < 0.0001), and the lesions treated in these patients were longer (*p* < 0.0001) and more frequently complex. Accordingly, the incidence of periprocedural myocardial infarction (*p* < 0.05) and that of cardiovascular death, target vessel and lesion failure and any PCI at 24 months was lower in the BVS-only group (all *p* < 0.05). The 24-months rate of definite and probable scaffold thrombosis was 2.7% in the hybrid group and 2.8% in the BVS-only group, that of stent thrombosis in the hybrid group was 1.86%. In multivariable analysis, only implantation in bifurcation lesions emerged as a predictor of device thrombosis, while the device type was not associated with this outcome (*p* = 0.21). The higher incidence of events in patients receiving hybrid stenting reflects the higher complexity of the lesions in these patients; in patients treated with a hybrid strategy, the type of device implanted did not influence patients´ outcomes.

## 1. Introduction

A number of randomized controlled trials comparing the outcomes of drug eluting stents compared to first-generation everolimus-eluting coronary bioresorbable vascular scaffolds (BVS) have shown the limitations of this novel type of devices [1,2,3,4]. When compared with drug eluting stents (DES), the mechanical limitations of BVS, including thicker and wider struts, lower radial strength, and limited expansion capabilities [5,6] represent important limitations for the treatment of complex lesions, including ostial or calcific ones, bifurcations, and lesions in small vessels. Supporting this concept, a number of post-hoc analyses have shown that this type of lesions represents predictors for BVS failure [7,8,9] unless a dedicated implantation technique is used [10,11]. Additionally, lesions in the left main, in by-pass grafts, and restenotic lesions have been excluded from the CE certification from the very beginning.

Based on these considerations, some authors have advocated for the use of a hybrid approach, which consists of limiting the use of BVS to settings in which the use of BVS is allowed (or considered to be safe) [12]. While this strategy is in conflict with the concept of “vascular regeneration” which represents the foundation of the use of BVS, it might still have the theoretical advantage that vessels (e.g., the proximal segments) in which long-term complications are clinically more relevant, would be “stent-free” after device resorption. Independently of the clinical rationale supporting the use of hybrid stenting, this setting however allows a direct head-to-head comparison of the outcomes of the device types independently of patients´ characteristics and clinical presentation.

The multicenter German-Austrian ABSORB Registry (GABI-R) was designed to monitor the usage of BVS in everyday practice. Details on this international registry have been published elsewhere [13]. In the current analysis, we set out to assess the incidence of clinical events in patients receiving hybrid percutaneous coronary interventions.

## 2. Methods

Between November 2013 and January 2016, consecutive patients undergoing implantation of at least one BVS (Absorb; Abbott Vascular, Santa Clara, CA, USA) were enrolled in a prospective single-arm registry in 92 GABI-R centers. Details on the methods for patients’ inclusion and follow-up in this observational registry have been previously published [13,14,15]. The study was conducted in accordance with the provisions of the Declaration of Helsinki and with the International Conference on Harmonization Good Clinical Practices, the protocol was approved by each local ethics committee (first Vote: Ethic committee of the Justus Liebig Universität Giessen 190/13) and all patients provided written, informed consent. Clinicaltrial.gov NCT02066623

### 2.1. Objective of the Study

The objective of this study was to investigate the outcome of patients receiving hybrid stenting with at least one drug eluting stent and one bioresorbable scaffold.

### 2.2. Procedures

Lesion preparation, BVS implantation, postdilation and use of intracoronary imaging, as well as medical therapy, were left to the operator’s discretion. The protocol recommended use of pre- and postdilation. High-pressure dilation was defined as dilation with ≥14ATM. Antiplatelet therapy consisted of aspirin (loading dose 250–500 mg and maintenance dose 100 mg/day) and clopidogrel (loading dose at least 300 mg and maintenance dose 75 mg/day), prasugrel (loading dose 60 mg and maintenance dose 10 mg), or ticagrelor (loading dose 180 mg and maintenance dose 90 mg bid). Dual antiplatelet therapy was recommended for at least 12 months.

### 2.3. Definitions

For the purpose of the present analysis, hybrid stenting was defined as implantation of at least one Absorb BVS and one metallic stent (BMS or DES) in the same patient. The primary endpoint of the present study was the incidence of definite/probable device thrombosis in lesions/patients treated with BVS compared to metallic stents.

Procedural success was defined as visually estimated residual stenosis <30% with thrombolysis in myocardial infarction flow grade III. Other definitions were based on the Academic Research Consortium (ARC) criteria [16]. Scaffold thrombosis was defined as definite or probable. Cardiac death was defined as death from immediate cardiac causes or complications related to the procedure as well as any death in which a cardiac cause could not be excluded. Myocardial infarction (MI) was defined according to the World Health Organization extended definition. Target lesion failure (TLF) was defined as a composite of cardiac death, target vessel MI, and clinically-driven target lesion revascularization (TLR). Target vessel failure (TVF) was defined as a composite of cardiac death, target-vessel MI, and clinically driven target vessel revascularization (TVR).

### 2.4. Data Management and Outcomes of Interest

Data in the GABI-R were collected electronically via an internet-based application and centralized by the IHF GmbH-Institut für Herzinfarktforschung (Ludwigshafen, Germany). Patients were contacted by telephone at 30 days, six months and two years using standardized questionnaires. Follow-up, source verification, quality controls were performed centrally. All events were adjudicated and classified by an independent event adjudication committee.

### 2.5. Statistical Analysis

Data are presented as mean ± standard deviation, absolute frequencies and percentages, or median (lower, upper quartile) as appropriate. Data are presented per patient and per lesion. Odds-ratios (95% confidence limits) are presented to characterize the differences in event frequencies among groups. The incidence of events in the periprocedural interval and at each of the follow-up times was tested with Pearson’s Chi-squared test. Concerning device thrombosis, testing for differences on patient level had to face a highly unbalanced design: There was no reference group for DES/BMS-only treatment. Thus, we implemented a loglinear model for an incomplete contingency table and three factors: BVS thrombosis, stent thrombosis, and hybrid treatment, accounting for interactions between the treatment and device type. To compare times to event (= device thrombosis) and assess the impact of the device type and hybrid stenting on outcomes, a proportional-hazard model (“Cox regression”) on stent level was implemented. Intra-subject correlations were considered by using a robust sandwich estimate aggregating stent residuals to subject level. This multiple regression model included the device type as a main factor and additional pre-defined predictor variables that have been previously shown to be associated with scaffold/stent thrombosis in the GABI-R: Total stent length, lesion type, bifurcation lesion, and time of implantation (before or after January 2015). Missing values were imputed either by random drawing from the standardized empirical distribution (in case of missing times-to-event), by modal values (binary) or by median values (metrical variables). A two-tailed *p* value <0.05 was considered to indicate statistical significance. Statistical analyses were performed using the SAS^®^ software, version 9.4 for Windows. Copyright © 2002–2012 SAS Institute Inc. SAS and all other SAS Institute Inc. product or service names are registered trademarks or trademarks of SAS Institute Inc., Cary, NC, USA.

## 3. Results

### 3.1. Patient Characteristics

CONSORT flow diagrams are presented in Figure 1 (left and right panel). Of 3144 (4278 lesions) patients included in the GABI-R registry who received at least one BVS and whose two-years vital status was known, 2709 (3308 lesions) were treated with scaffolds only (BVS-only group) while 435 (970 lesions) were treated with at least one additional metallic stent (hybrid group).

Patient characteristics are presented in Table 1. Patients in the hybrid group consistently showed characteristics compatible with a higher complexity: Glomerular filtration rate was lower (*p* < 0.05), the prevalence of prior PCI (*p* < 0.01), myocardial infarction (*p* < 0.01), multivessel disease (*p* < 0.0001), male sex (*p* < 0.05) were all higher in the hybrid group and there was a trend towards older age and higher diabetes prevalence in this group (both = 0.06). In line with this, procedure duration, contrast use, radiation time, and the number of lesions treated per patient were larger in the hybrid group (all *p* < 0.0001). DAPT with prasugrel was used more commonly in the hybrid group (*p* < 0.05). The prevalence of smoking was higher in the BVS-only group (*p* < 0.05).

Lesion characteristics are presented in Table 2. A total of 4962 BVS/Stents (4349 BVS, 631 in the hybrid group and 3718 in the BVS-only group, and 613 stents, all in the hybrid group) were implanted. The large majority of metallic stents were DES (total of DES used: 610), and only three BMS were used. Interventions in the hybrid group were more frequent in the LAD, those in the BVS-only group were more frequent in the RCA (*p* < 0.0001 and *p* < 0.05). Compatible with the above differences between groups, all parameters expressing lesion complexity were more frequent in the hybrid group: The prevalence of B2 (*p* < 0.05), C1 (*p* < 0.0001), C2 (*p* < 0.05) lesions, bifurcation lesions (*p* < 0.0001), chronic total occlusions (*p* < 0.0001), lesions with severe tortuosity (*p* < 0.05), presence of calcium (*p* < 0.05), and lesion length (*p* < 0.0001) were higher in the hybrid group than in the BVS only group. Predilatation was performed in 93.5% of BVS-only treated patients and 85.7% of patients treated with hybrid-PCI (*p* < 0.0001). The use of high-pressure inflations, scoring balloons, rotablator, was more frequent in the hybrid group (*p* < 0.001). In contrast, postdilation was performed more frequently (73.6% compared to 68.4% in BVS-only patients.

Lesion and procedural characteristics in the hybrid group are presented in Table 3. Of the 970 lesions in patients in the hybrid group, 417 (43.8%) had been treated with BVS only, 410 (43.1%) with DES/BMS only and there was a total of 124 lesions treated with overlapping hybrid strategy (2.9% of the total, 12.8% of the lesions treated in patients who received hybrid revascularization). An additional 19 were not classified in the database. When lesions treated with BVS-only were compared to lesions treated with DES/BMS only, BVS-only lesions were longer, more frequently type C2 (both *p* < 0.05), and there was a trend towards more frequent chronic total occlusions (*p* = 0.06). Only the prevalence of bifurcation lesions was higher in the DES/BMS-treated lesions (*p* < 0.0001). There was a total of 25 Medina 1,1,1 lesions, and 2 Medina 0,1,1 lesions in the hybrid group. There were only three cases of hybrid bifurcation stenting (metallic stent + BVS in the same bifurcation lesion). In terms of procedural parameters, larger predilation balloons, imaging and postdilation were used more frequently in BVS-treated lesions (*p* < 0.01, *p* < 0.05 and *p* < 0.0001). Procedural success was 99% in both groups.

### 3.2. Clinical Outcomes

The incidence of periprocedural myocardial infarction (*p* < 0.05, OR 4.2(1.2–14.9)) and vessel perforation (*p* < 0.001, OR 4.9(1.8–13.2)) was higher in the hybrid group. Otherwise, there was no difference in the incidence of periprocedural events.

At 30 days (Table 4), the incidence of cardiovascular death, target vessel and target lesion failure were higher in the hybrid group. Similarly, at 24-month follow-up (follow up available in 98.4% of the patients), the incidence of cardiovascular death (*p* < 0.05, OR 2.3(1.0–5.2)), target vessel failure (*p* < 0.01, OR 1.7(1.2–2.3)) and target lesion failure (*p* < 0.05, OR 1.6(1.1–2.3)), and that of any PCI (*p* < 0.05, OR 1.4(1.1–1.8)), were higher in the hybrid group.

There was no significant difference (*p* = 0.13) in the incidence of target lesion revascularization (estimates and confidence limits presented in Figure 2).

A total of 17 definite/probable stent thromboses occurred in the hybrid group during the 24-months follow-up: In six cases, they affected both (at least) a DES and a BVS in the same patient; in four cases, they only affected a BVS, and in one case only one DES.

Figure 3A,B show the two-years incidence, as well as estimates and confidence limits for the incidence of stent and BVS thrombosis in both the hybrid and BVS-only group. Before testing for differences, effects of the device type and hybrid treatment had to be separated and adjusted for possible interactions. Thereafter, with regard to treatment strategy, only a trend towards a higher incidence of BVS thrombosis remained in the BVS-only group (*p* = 0.07). With regards to the device type, BVS and stent thrombosis rates did not differ significantly, neither within the hybrid group (*p* = 0.22), between hybrid and BVS-only group (*p* = 0.31), nor pooled over all treatments (*p* = 0.07).

In the multivariable analysis, only the implantation in bifurcation lesions emerged as an independent predictor of device thrombosis (Table 5). In separate analyses neither acute coronary syndrome at index nor the implantatation technique used modified this association (*p* = 0.241 and *p* = 0.637, respectively).

## 4. Discussion

The GABI-R is a large international registry on the use of BVS. In the present analysis, we investigate the characteristics and outcomes of patients who received hybrid stenting (i.e., at least one BVS and at least one metallic stent). The major findings of the current analysis include: (i) Patients treated with hybrid stent/scaffold therapy had a more complex presentation and worse outcomes than those treated with BVS alone; (ii) During a two-years follow-up, the incidence of adverse events at the level of the lesions treated with BVS was not worse than that of the lesions treated with metallic stents.

The concept of a vascular scaffold that provides temporary mechanical support and is resorbed during follow-up to avoid a permanent unnecessary foreign body remains an attractive concept for percutaneous coronary intervention. Although initial randomized trials reported non-inferiority as compared to metallic drug-eluting stents [17,18], with signals that these devices might also be used in more complex lesion [19,20,21,22], more recent trials have consistently demonstrated a higher incidence of adverse events both early and late after implantation [3,4,23]. Mechanistic evidence shows that these increased rates might depend on the technique used at the time of implantation, inadequate selection of lesions, and, importantly, on the mechanical limitations of the devices, including increased strut thickness, reduced expansion limits, reduced radial resistance [6,8]. Further, particular settings, such as chronic total occlusions, acute coronary syndromes, treatment of ostial lesions and a lack of care at the time of implantation have all been associated with increased events, including thrombosis and restenosis [3,24,25,26,27,28]. Based on these notions, the general recommendation is that patients who have been treated with BVS should prolong their dual antiplatelet therapy until complete resorption of the device. This recommendation is likely to be particularly important in patients treated for complex lesions, such as those presented here. Knowledge of these limitations lead to the hypothesis that, by avoiding implantation in the presence of adverse lesion characteristics (e.g., long lesions with proximal or distal reference vessel diameters unsuitable for BVSs, very calcific or bifurcation lesions) may represent an adequate compromise between the benefit of the BVS and the risk of adverse events. Interestingly, this setting also allows within-patients comparison of the outcomes, i.e., removes the confounding influence of differences in patient-related risk factors among groups.

Beyond any consideration on the safety of the devices implanted, it might be hypothesized that the use of BVS might be more advantageous in long and proximal segments, which might thereafter regain the possibility to adjust their diameter in response to biochemical and physical stimuli. Calcific lesions, in contrast, might have theoretically less potential for regeneration. Thrombotic lesions might also represent a setting for BVS, allowing ”plaque stabilization“ as previously reported [29]. The use of BVS in CTO lesions has also been reported, but no data are available regarding the capacity of these lesions to regenerate [30]. In the present database, lesions treated with BVS-only were indeed longer and more frequently of type C (thrombotic). There was a trend towards less calcific lesions being treated with BVS-only, but this difference remains speculative. Finally, in theory there is also a rationale for the use of BVS in ostial or bifurcation lesions to limit (in time) the risks associated with malapposed struts, but the evidence on their (lack of) safety in these settings clearly discouraged their use [31]. Based on the instructions for use, in the present database, bifurcation lesions were almost exclusively treated with metallic stents.

Reports on the short-term outcomes of this so-called hybrid stenting strategy (the use of both metallic stents and BVS in the same patient or lesion) have been previously published [32,33,34,35]. Collectively, these studies reported that the use of a hybrid approach might be an acceptable compromise to overcome the limitations of BVS. In the present study, we report data from a larger database with longer follow-up. In our study, the incidence of events was similar between BVS and metallic stents, while treatment of complex patients and lesions (particularly bifurcation lesions) remained an independent predictor of events. These findings confirm the hypothesis that a hybrid approach, in which more complex settings are treated with DES, might be a feasible option, although its rationale needs to be validated. While polymeric devices of the first generation (Absorb, Abbott vascular) have been removed from the market following the evidence of increased adverse events, the present results might also apply to other similar devices for which data from large databases are not available. Further, they provide a perspective for novel devices of this type.

## 5. Limitations

The GABI-R was a prospective registry designed to provide information on the real-life use of BVS and is therefore affected by the limitations of this type of study design. Centralized data monitoring, quality assessment, and follow-up were however performed to limit these issues. With regards to the present analysis, hybrid treatment of lesions complicates the adjudications of the events to one or the other device type. For this reason, the comparison of the incidence of device thrombosis was limited to lesions treated with only one type of device. Despite the size of the database, conclusions on very rare subsets (e.g., bifurcation lesions treated with hybrid strategy) were not possible. As well, data on the antiplatelet regimen at the time of the event are missing. The present data should not be directly extrapolated to second-generation BVS. However, they provide an insight that a prudent strategy of hybrid stenting might allow combining the benefit of bioresorbable devices with the safety of standard metallic stents also in more complex settings. The impact of a correct implantation technique has been demonstrated in a number of papers, including those from our group [8,18,24,25,27,28]. Unfortunately, the absence of a central quantitative coronary analysis in the present database does not allow clear conclusions to this regard. Finally, the comparison of the outcomes within hybrid patients removes patient-related confounders but not lesion-related confounders, which would be better addressed in trials with a randomization at lesion level.

## 6. Conclusions

We report on the outcome of patients undergoing BVS and DES implantation, a particularly complex subset among patients treated with BVS. In this database, which is one of the largest ones worldwide on the use of BVS, the type of device implanted did not influence patients´ outcomes. Hybrid stenting is a negotiation between the concept of “full vascular regeneration” and the mechanical limitations of these novel devices. Whether the use of metallic stents, although limited as compared to a full-metal strategy, compromises the benefits of BVS remains however to be discussed. Whether the use of a hybrid strategy with newer (and safer) scaffolds will present any advantage as compared to a full-DES strategy, will need to be studied in the future.

## Figures and Tables

**Figure 1 jcm-08-00767-f001:**
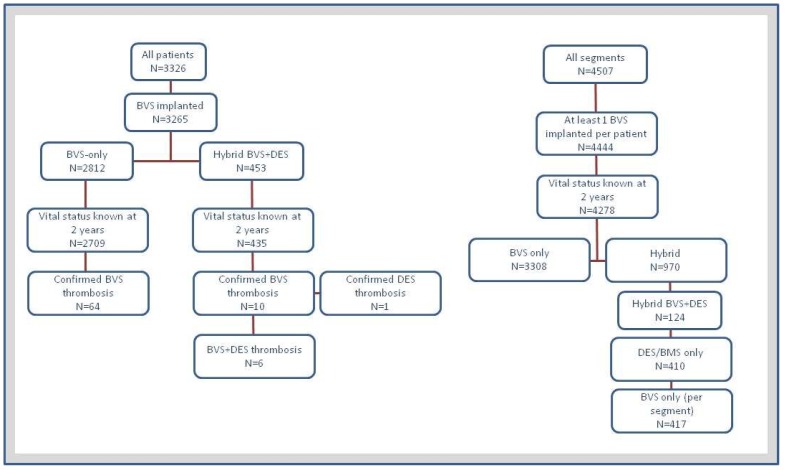
Study flow per patient (**left** panel) and per lesion (**right**).

**Figure 2 jcm-08-00767-f002:**
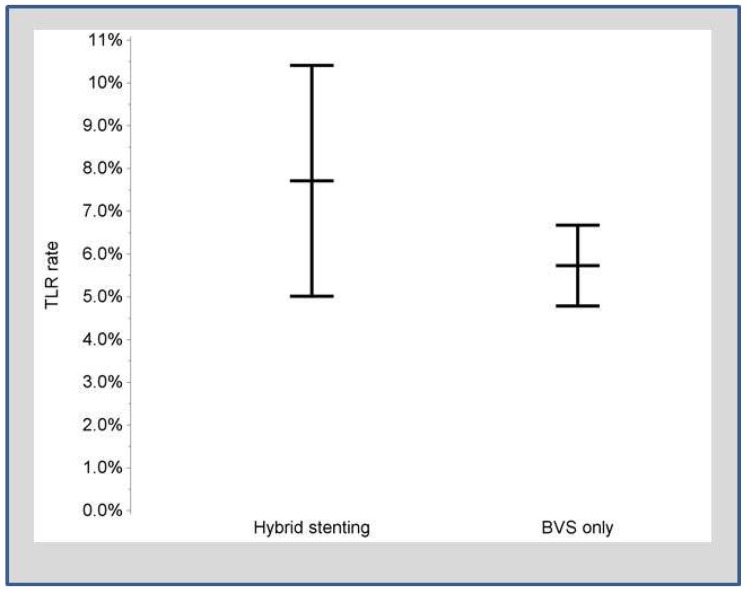
Two-years rates of target lesion revascularization. Comparison of Patients treated with hybrid vs. BVS-only strategy.

**Figure 3 jcm-08-00767-f003:**
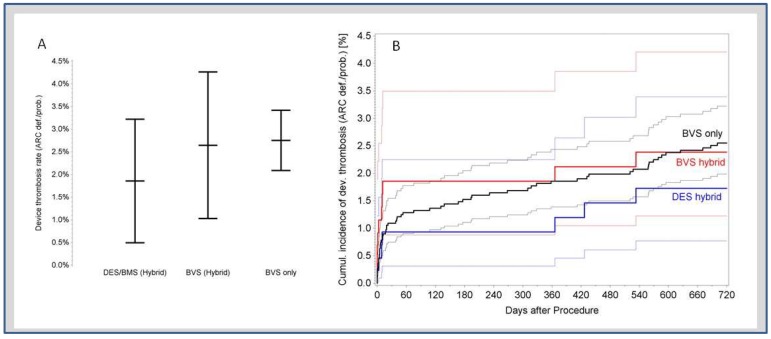
(**A**) Two-years incidence of device thrombosis. There was no difference among lesions treated with BVS only, metal stents only, or hybrid strategies; (**B**) cumulative incidence curves showing an overlap of the confidence intervals.

**Table 1 jcm-08-00767-t001:** Baseline characteristics of the cohort.

	Total (Hybrid + BVS-Only) *n* = 3144	Hybrid Group *n* = 435	BVS Only *n* = 2709	*p* Value
Female gender	22.9% (721/3144)	19.1% (83/435)	23.6% (638/2709)	<0.05
Age (years, rounded)	60.87 ± 11.02	61.91 ± 10.36,	60.7 ± 11.11	0.06
Diabetes mellitus	20.9% (651/3117)	24.2% (105/433)	20.3% (546/2684)	0.06
Current smoker	34.9% (1039/2978)	30.4% (128/421)	35.6 % (911/2557)	<0.05
Arterial hypertension	73.4% (2274/3100)	75.3% (324/430)	73% (1950/2670)	0.31
Hypercholesterolemia	56.5% (1702/3010)	59.6% (243/408)	56.1% (1459/2602)	0.19
Glomerular filtration rate	79.39 ± 23.68, *n* = 1590	75.33 ± 22.55, *n* = 165	79.86 ± 23.77, *n* = 1425	<0.05
History of myocardial infarction	22.2% (687/3094)	27.4% (117/427)	21.4% (570/2667)	<0.01
History of PCI	33.9% (1044/3079)	39.6% (169/427)	33% (875/2652)	<0.01
History of aorto-coronary bypass surgery	2.5% (79/3131)	3% (13/433)	2.4% (66/2698)	0.49
History of CAD	41.1% (1137/2768)	44% (178/405)	40.6% (959/2363)	0.20
History of stroke	2.7% (85/3143)	3% (13/435)	2.7% (72/2708)	0.69
Acute coronary syndrome at presentation	51.4% (1617/3143)	47.8% (208/435)	52% (1409/2708)	0.10
Stable angina pectoris	33.5% (1053/3143)	33.1% (144/435)	33.6% (909/2708)	0.85
Left ventricular ejection fraction	56.09 ± 10.5, *n* = 1930	54.84 ± 10.15, *n* = 282	56.31 ± 10.55, *n* = 1648	<0.05
1-vessel CAD	41.9% (1317/3144)	20% (87/435)	45.4% (1230/2709)	<0.0001
2-vessels CAD	31% (974/3144)	35.6% (155/435)	30.2% (819/2709)	<0.05
3-vessel CAD	27.1% (852/3144)	44.4% (193/435)	24.3% (659/2709)	<0.0001

Values are mean ± SD or % (absolute number/number of available records); CAD = coronary artery disease; PCI = percutaneous coronary intervention, CBR = clinical BVS restenosis.

**Table 2 jcm-08-00767-t002:** Angiographic and procedural characteristics.

	Total (Hybrid + BVS-Only)	Hybrid Group	BVS Only	*p* Value
Procedure duration, minutes	58.90 ± 28.91, *n* = 3141	77.83 ± 35.96, *n* = 435	55.85 ± 26.38, *n* = 2706	<0.0001
Radiation time, minutes	11.84 ± 8.22, *n* = 3143	18.05 ± 10.64, *n* = 435	10.84 ± 7.28, *n* = 2708	<0.0001
Amount of contrast medium, mL	174.76 ± 74.65, *n* = 3140	223.40 ± 91.21, *n* = 435	166.94 ± 68.50, *n* = 2705	<0.0001
IVUS	3% (94/3142)	2.8% (12/435)	3% (82/2707)	0.76
OCT	4.5% (141/3142)	4.6% (20/435)	4.5% (121/2707)	0.90
Per lesion				
Treated segments	4278	970	3308	
Lesions treated with BRS only	87.3% (3670/4204)	43.8% (417/951)	100% (3253/3253)	
Lesions treated with stents only	9.8% (410/4204)	43.1% (410/951)	0% (0/3253)	
Intervention in LAD	74.7% (1582/2118)	84.7% (287/339)	72.8% (1295/1779)	<0.0001
Intervention in LCX	59.6% (813/1363)	63.8% (153/240)	58.8% (660/1123)	0.15
Intervention in RCA	68.4% (1051/1537)	62.8% (155/247)	69.5% (896/1290)	<0.05
Graft	23.8% (5/21)	0% (0/1)	25% (5/20)	0.57
Lesion type				
A	26.5% (1133/4270)	19.4% (187/964)	28.6% (946/3306)	<0.0001
B1	37% (1579/4270)	36.1% (348/964)	37.2% (1231/3306)	0.52
B2	19.6% (836/4270)	21.9% (211/964)	18.9% (625/3306)	<0.05
C1	12.6% (539/4270)	17% (164/964)	11.3% (375/3306)	<0.0001
C2	4.3% (183/4270)	5.6% (54/964)	3.9% (129/3306)	<0.05
De novo lesion	94.2% (4025/4272)	92.9% (897/966)	94.6% (3128/3306)	<0.05
Ostial lesion	0.8% (36/4272)	0.5% (5/966)	0.9% (31/3306)	0.21
Bifurcation lesion	2.9% (123/4272)	5.5% (53/966)	2.1% (70/3306)	<0.0001
100% stenosis	5.6% (241/4272)	5.7% (55/966)	5.6% (186/3306)	0.94
Chronic total occlusion	37.3% (90/241)	63.6% (35/55)	29.6% (55/186)	<0.0001
Severe tortuosity	1.2% (52/4263)	1.9% (18/961)	1% (34/3302)	<0.05
No calcification	35.9% (1533/4270)	33.2% (320/964)	36.7% (1213/3306)	<0.05
% Stenosis	86.30 ± 11.73, *n* = 4275	84.92 ± 11.93, *n* = 968	86.71 ± 11.65, *n* = 3307	<0.0001
Imaging	3.2% (136/4275)	1.8% (17/968)	3.6% (119/3307)	<0.01
FFR	5.2% (223/4262)	6.7% (64/961)	4.8% (159/3301)	<0.05
RVD	2.95 ± 0.63, *n* = 93	3.15 ± 0.43, *n* = 11	2.92 ± 0.64, *n* = 82	0.26
Lesion length	17.12 ± 9.30, *n* = 4258	18.84 ± 10.51, *n* = 956	16.62 ± 8.85, *n* = 3302	<0.0001
Lesion length >34 mm	5.6% (238/4258)	8.4% (80/956)	4.8% (158/3302)	<0.0001
Any lesion preparation	91.7% (3921/4274)	85.7% (830/968)	93.5% (3091/3306)	<0.0001
Pre-dilatation	100% (3920/3921)	100% (830/830)	100% (3090/3091)	0.60
High pressure balloon	43% (1680/3908)	49.3% (408/828)	41.3% (1272/3080)	<0.0001
Non-compliant balloon	73% (1215/1665)	85.3% (348/408)	69% (867/1257)	<0.0001
Use of scoring balloon	3% (116/3921)	5.4% (45/830)	2.3% (71/3091)	<0.0001
Rotablation	0.2% (6/3921)	0.6% (5/830)	0% (1/3091)	<0.001
Stent/BVS size, mm	3.07 ± 0.59, *n* = 4960	3.03 ± 0.45, *n* = 1243	3.08 ± 0.63, *n* = 3717	<0.001
Postdilatation performed	72.4% (3093/4271)	68.4% (660/965)	73.6% (2433/3306)	<0.01
High-pressure Postdilation	89.5% (2766/3090)	86.9% (573/659)	90.2% (2193/2431)	<0.05
PSP-technique	6.4% (244/3794)	12.6% (68/541)	5.4% (176/3253)	<0.0001
Procedural success	99% (4229/4273)	98.7% (954/967)	99.1% (3275/3306)	0.27
Glycoprotein IIb/IIIa inhibitors	8% (252/3143)	6.7% (29/435)	8.2% (223/2708)	0.26
**Medical therapy at discharge**				
Aspirin	97.3% (3056/3141)	95.9% (417/435)	97.5% (2639/2706)	<0.05
P2Y12-receptor inhibitorsClopidogrel	44% (1351/3068)	41.2% (175/425)	44.5% (1176/2643)	0.2
Prasugrel	34.1% (1045/3068)	38.6% (164/425)	33.3% (881/2643)	<0.05
Ticagrelor	21.9% (672/3068)	20.2% (86/425)	22.2% (586/2643)	0.37

Values are mean ± SD, median (quartiles) or % (absolute number/number of available records); BVS = bioresorbable vascular scaffold; CBR = clinical BVS restenosis; DES = drug eluting stent; PCI = percutaneous coronary intervention.

**Table 3 jcm-08-00767-t003:** Lesion-level analysis, bioresorbable vascular scaffold (BVS)-treated lesions compared to lesions treated with metallic stents in the hybrid group.

	Total	BVS Only	DES/BMS Stent Only	*p* Value	OR (95%-CI)
Number of lesions	827	417	410		
Stenosis (%) before PCI	84.42 ± 11.94, *n* = 827	84.47 ± 11.36, *n* = 417	84.37 ± 12.51, *n* = 410	0.69	
RVD (mm)	2.96 ± 0.29, *n* = 6	2.96 ± 0.24, *n* = 3	2.95 ± 0.39, *n* = 3	1	
Lesion length (mm)	18.01 ± 9.9, *n* = 815	18.6 ± 9.66, *n* = 411	17.41 ± 10.1, *n* = 404	<0.05	
Lesion length >34 mm	6.5 % (53/815)	6.8% (28/411)	6.2% (25/404)	0.72	1.11 (0.63–1.94)
**Morphology**					
A	20.4% (168/823)	21.4% (89/416)	19.4% (79/407)	0.48	1.13 (0.80–1.59)
B1	36.2% (298/823)	36.8% (153/416)	35.6% (145/407)	0.73	1.05 (0.79–1.40)
B2	22.6% (186/823)	20% (83/416)	25.3% (103/407)	0.07	0.74 (0.53–1.02)
C1	15.6% (128/823)	15.1% (63/416)	16% (65/407)	0.74	0.94 (0.64–1.37)
C2	5.2% (43/823)	6.7% (28/416)	3.7% (15/407)	<0.05	1.89 (0.99–3.59)
De novo vessel	93% (767/825)	93.8% (391/417)	92.2% (376/408)	0.37	1.28 (0.75–2.19)
In-stent re-stenosis	1% (8/825)	0.5 % (2/417)	1.5% (6/408)	0.15	0.32 (0.06–1.61)
Bifurcation	5.9% (49/825)	2.4% (10/417)	9.6% (39/408)	<0.0001	0.23 (0.11–0.47)
Complete occlusion	5.3% (44/825)	6.2% (26/417)	4.4% (18/408)	0.24	1.44 (0.78–2.67)
CTO	61.4% (27/44)	73.1% (19/26)	44.4% (8/18)	0.06	3.39 (0.95–12.09)
Ostial lesion	0.6% (5/825)	0.2% (1/417)	1% (4/408)	0.17	0.24 (0.03–2.18)
Severe tortuosity	2% (16/820)	1.2% (5/416)	2.7% (11/404)	0.12	0.43 (0.15–1.26)
No calcification	33.7% (277/823)	36.3% (151/416)	31% (126/407)	0.11	1.27 (0.95–1.7)
Mild	43.7% (360/823)	44.2% (184/416)	43.2% (176/407)	0.78	1.04 (0.79–1.37)
Moderate	18.2% (150/823)	15.9% (66/416)	20.6% (84/407)	0.08	0.73 (0.51–1.04)
Severe	4.4% (36/823)	3.6% (15/416)	5.2% (21/407)	0.28	0.69 (0.35–1.35)
**Procedural Characteristics**					
Pre-dilatation	100% (693/693)	100% (396/396)	100% (297/297)	n.d.	
High pressure balloon	51.1% (353/691)	52.3% (207/396)	49.5% (146/295)	0.47	1.12 (0.83–1.51)
Maximum balloon diameter (mm)	2.75 ± 0.46, *n* = 689	2.79 ± 0.41, *n* = 395	2.69 ± 0.5, *n* = 294	<0.01	
Scoring balloon	5.8% (40/693)	6.8% (27/396)	4.4 % (13/297)	0.17	1.6 (0.81–3.15)
Rotablation	0.6% (4/693)	0.5% (2/396)	0.7% (2/297)	0.77	0.75 (0.10–5.35)
Post-dilatation	66.8% (551/825)	85.5% (355/415)	47.8% (196/410)	<0.0001	
High pressure balloon	86.5% (476/550)	89.9% (319/355)	80.5% (157/195)	<0.01	

Intravasc. imaging (IVUS/OCT/QCA) after PCI	1.6% (13/827)	2.6% (11/417)	0.5% (2/410)	<0.05	
Procedural success	99% (819/827)	99% (413/417)	99% (406/410)	0.98	

Values are mean ± SD or % (absolute number/number of available records).

**Table 4 jcm-08-00767-t004:** Clinical Outcomes.

	Total (*n* = 3144)	Hybrid Stenting (*n* = 435)	BVS Only (*n* = 2709)	*p* Value	OR (95%-CI)
**Periprocedural complications**					
Death	0% (0/3143)	0% (0/435)	0% (0/2708)	n.d.	-
MI	0.3% (10/3143)	0.9% (4/435)	0.2% (6/2708)	<0.05	4.18 (1.17–14.87)
CABG - emergency operation	0% (0/3143)	0% (0/435)	0% (0/2708)	n.d.	-
Coronary thrombosis	0.4% (12/3143)	0.9% (4/435)	0.3% (8/2708)	0.05	3.13 (0.94–10.45)
Coronary perforation	0.5% (16/3140)	1.6% (7/435)	0.3% (9/2705)	<0.001	4.9 (1.82–13.22)
**30-days follow-up**					
All-cause mortality	0.51% (16/3144)	1.15% (5/435)	0.41% (11/2709)	<0.05	2.85 (0.99–8.25)
Cardiovascular mortality	0.32% (10/3144)	0.92% (4/435)	0.22% (6/2709)	< 0.05	4.18 (1.18–14.88)
Scaffold thrombosis Definite	0.86% (27/3144)	1.15% (5/435)	0.81% (22/2709)	0.48	1.42 (0.53–3.77)
- Probable	0.35% (11/3144)	0.69% (3/435)	0.3% (8/2709)	0.20	2.34 (0.62–8.87)
Stent thrombosis Definite	0.23% (1/435)	0.23% (1/435)		-	-
- Probable	0.69% (3/435)	0.69% (3/435)		-	-
Any myocardial infarction	1.43% (45/3144)	1.84% (8/435)	1.37% (37/2709)	0.44	1.35 (0.63–2.93)
Target vessel related MI	1.18% (37/3144)	1.61% (7/435)	1.11% (30/2709)	0.37	1.46 (0.64–3.35)
Target lesion revascularization	1.08% (34/3144)	1.38% (6/435)	1.03% (28/2709)	0.52	1.34 (0.55–3.25)
Target lesion failure	1.49% (47/3144)	2.76% (12/435)	1.29% (35/2709)	<0.05	2.17 (1.12–4.21)
Target vessel failure	1.72% (54/3144)	2.99% (13/435)	1.51% (41/2709)	<0.05	2 (1.07–3.77)
**24-months follow-up**					
Follow-up available	98.4% (3094/3144)	97.2% (423/435)	98.6% (2671/2709)		
All-cause mortality	3.06% (96/3135)	4.37% (19/435)	2.85% (77/2700)	0.09	1.56 (0.93–2.6)
Cardiovascular mortality	0.96% (30/3135)	1.84% (8/435)	0.81% (22/2700)	<0.05	2.28 (1.01–5.16)
Scaffold thrombosis Definite	2% (54/2694)	1.33% (5/375)	2.11% (49/2319)	0.32	0.63 (0.25–1.58)
- Probable	0.78% (21/2688)	1.33% (5/377)	0.69% (16/2311)	0.19	1.93 (0.7–5.29)
Stent thrombosis Definite	0.53% (2/374)	0.53% (2/374)		-	-
- Probable	1.33% (5/377)	1.33% (5/377)		-	-
Any myocardial infarction	5.07% (137/2703)	5.31% (20/377)	5.03% (117/2326)	0.82	1.06 (0.65–1.72)
Target vessel related MI	3.37% (91/2700)	3.19% (12/376)	3.4% (79/2324)	0.84	0.94 (0.51–1.74)
Target lesion revascularization	6% (162/2698)	7.71% (29/376)	5.73% (133/2322)	0.13	1.38 (0.91–2.09)
Target lesion failure	7.19% (195/2711)	10.24% (39/381)	6.7% (156/2330)	<0.05	1.59 (1.1–2.3)
Target vessel failure	10.21% (277/2714)	14.7% (56/381)	9.47% (221/2333)	<0.01	1.65 (1.2–2.26)
Any PCI	18.52% (505/2727)	23.02% (87/378)	17.79% (418/2349)	<0.05	1.38 (1.06–1.79)

Values are mean ± SD or % (absolute number/number of available records); BRS = bioresorbable vascular scaffold; CI = confidence interval; PCI = percutaneous coronary intervention; OR = Odds ratio.

**Table 5 jcm-08-00767-t005:** Multivariate analysis of the predictors of definite/probable device thrombosis.

Analysis of Maximum Likelihood Estimates					
Parameter	Parameter Estimate	Standard Error	Chi-Square	Pr > ChiSq	Hazard Ratio
**Device type**	0.44883	0.36030	1.5518	0.2129	1.566
**Total stent length**	0.21102	0.16178	1.7015	0.1921	1.235
**Lesion type B2/C**	−0.32342	0.26101	1.5354	0.2153	0.724
**Implantation after Jan. 2015**	−0.33539	0.23905	1.9683	0.1606	0.715
**Bifurcation**	1.04114	0.48421	4.6233	0.0315	2.832

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
