# Peer review of "Hybrid Coronary Percutaneous Treatment with Metallic Stents and Everolimus-Eluting Bioresorbable Vascular Scaffolds: 2-Years Results from the GABI-R Registry"

_jcm, 2019, doi:10.3390/jcm8060767_

Round 1
Reviewer 1 Report
In the manuscript entitled "Hybrid Coronary Percutaneous Treatment with Metallic Stents and Everolimus-Eluting Bioresorbable Vascular Scaffolds: 2-Years Results from the GABI-R Registry" authors lead by Tommaso Gori present a subgroup analysis of the GABI-R Registry focused on the analysis of outcomes of patients receiving hybrid stenting with both BVS and metallic stent compared to patients that receiving BVS only.
The authors should be commended in their effort, as this analysis could give further insights for future considerations on the BVS application. Despite this technology is no longer available and this may limit is clinical impact, results of this analysis may be important to understand the clinical implications in the patients in which this procedure was already done and to set the stage for future validation of these findings when novel iteration of the technology will be available.
This reviewer has the following comments:
· Are there any information regarding patients undergoing hybrid strategy in the same vessel ? (overlapping vs. non overlapping hybrid stenting)
· Bifurcation stenting: was this made routinely with the same stent or are there cases with hybrid (BVS/DES) bifurcation stenting ?
· Are information regarding DAPT type and duration available in the dataset? Did the Authors include this information in the multivariable model for stent thrombosis ?
· The authors should provide also an outlook about the possible antithrombotic strategies in these patients with complex PCI in the discussion section of the manuscript.
Please also consider these minor comments:
· At line 329 “de Ribamar Costa J, Jr.,” should be “de Ribamar Costa JJ,”.
· At line 38 “+” before “probable” should be deleted.
Author Response
Reviewer 1
[...]The authors should be commended in their effort, as this analysis could give further insights for future considerations on the BVS application. Despite this technology is no longer available and this may limit is clinical impact, results of this analysis may be important to understand the clinical implications in the patients in which this procedure was already done and to set the stage for future validation of these findings when novel iteration of the technology will be available.
Thank you very much for your review and comments.
This reviewer has the following comments:
· Are there any information regarding patients undergoing hybrid strategy in the same vessel ? (overlapping vs. non overlapping hybrid stenting)
Yes, this information is available.
These sentences have been added to the results section:
There was a total of 124 lesions treated with overlapping hybrid strategy (2.9% of the total, 12.8% of the lesions treated in patients who received hybrid revascularization).
Given the relatively small numbers, it is impossible to conclude on the outcome of these patients/lesion specifically.
· Bifurcation stenting: was this made routinely with the same stent or are there cases with hybrid (BVS/DES) bifurcation stenting ?
There was a total of 25 Medina 1,1,1 lesions, and 2 Medina 0,1,1 lesions in the hybrid group. There were only three cases of hybrid bifurcation stenting, the largest majority of these lesions were treated with metallic stents. This information is now reported in the results section, page 9 and 10.
· Are information regarding DAPT type and duration available in the dataset? Did the Authors include this information in the multivariable model for stent thrombosis ?
This is a very important question. A comment on the type of DAPT recommended have been added (line 154, results), the data are reported in Table 2. There was a slightly larger use of prasugrel in the hybrid group. Following the expert consensus, 12 months DAPT were recommended. The data on „DAPT at the time of event“ are unfortunately very sparse and unreliable in the database. This has been acknowledged as limitation.
· The authors should provide also an outlook about the possible antithrombotic strategies in these patients with complex PCI in the discussion section of the manuscript.
This is an important point. A sentence to this regard has been added:
„[...]Based on these notions, the general recommendation is that patients who have been treated with BVS should prolong their dual antiplatelet therapy until complete resorption of the device. This recommendation is likely to be particularly important in patients treated for complex lesions, such as those presented here.“.
Please also consider these minor comments:
· At line 329 “de Ribamar Costa J, Jr.,” should be “de Ribamar Costa JJ,”.
· At line 38 “+” before “probable” should be deleted.
Thank you very much, these errors have been corrected.
Reviewer 2 Report
Little data are available regarding the outcomes of patients receiving hybrid stenting with both bioresorbable scaffolds (BVS) and drug-eluting stents (DES). In the GABI-Registry, clinical outcomes were compared between hybrid group (435 patients, 970 lesions) and BVS-only group (2709 patients, 3308 lesions) who received BVS-only. The incidence of peri-procedural myocardial infarction and that of cardiovascular death, target vessel and lesion failure and any PCI at 24 months was lower in the BVS-only group. The 24-months rate of definite + probable scaffold thrombosis was 2.7% in the hybrid group and 2.8% in the BVS-only group. In multivariable analysis, only implantation in bifurcation lesions emerged as predictor of device thrombosis, while device type was not associated with this outcome. They concluded that the higher incidence of events in patients receiving hybrid stenting reflects the higher complexity of the lesions in these patients; in patients treated with a hybrid strategy, the type of device implanted did not influence patients´ outcomes.
Although this paper is very interesting, I suggest some revisions.
1. First of all, the strong point of hybrid strategy is unclear. It may be beneficial regarding the long-term outcomes to treat all lesions by BVS because they are resorbed in the future. However, if even one lesion are treated by metallic stent, the merit of the use of BVS for the other lesions would not be big. Authors must mention the stronger point of hybrid strategy compared to DES-only strategy in the background.
2. They did not mention the clinical implication enough. We would like to know which kind of lesions are appropriate for hybrid strategy or BVS-only strategy. They must analyze and mention them in the result and discussion sections
Author Response
Reviewer 2
We are grateful for your comments, and hope that our responses are satisfactory.
1. First of all, the strong point of hybrid strategy is unclear. It may be beneficial regarding the long-term outcomes to treat all lesions by BVS because they are resorbed in the future. However, if even one lesion are treated by metallic stent, the merit of the use of BVS for the other lesions would not be big. Authors must mention the stronger point of hybrid strategy compared to DES-only strategy in the background.
You are absolutely right. We defintely do not want to advocate for a strategy where BRS are reserved for „easy“, type A, lesions, while DES are used for all other lesion types. The rates of complications after implantation of any device in type A lesions is so low, that any new device must show (at least long-term) superiority in complex setting. The outcomes after use of a hybrid strategy deserves investigation, but, given the results of BVS studies, should definitely not be recommended.
On the other side, BRS of the Absorb 1.1 generation were still first-generation devices, with important mechanical limitations that were known from the beginning and additional ones that became more evident later. The rationale for hybrid stenting at that time was that „critical“ (e.g. proximal LAD, LCX and RCA) segments would still profit from BVS, while the difference between devices would be smaller when used in smaller (clinically less relevant) segments. We agree with you that this concept remains to be demonstrated.
We have added two sentences to clarify this:
Introduction:
Additionally, lesions in the left main, in by-pass grafts, and restenotic lesions have been excluded from the CE certification from the very beginning.
Based on these considerations, some authors have advocated for the use of a hybrid approach, which consists of limiting the use of BVS to settings in which the use of BVS is allowed (or considered to be safe)1. While this strategy is in conflict with the concept of “vascular regeneration” which represents the foundation of the use of BVS, it might still have the theoretical advantage that vessels (e.g., the proximal segments) in which long-term complications are clinically more relevant, would be “stent-free” after device resorption. Independently of the clinical rationale supporting the use of hybrid stenting, this setting however allows a direct head-to-head comparison of the outcomes of the device types independently of patients´characteristics and clinical presentation.
Conclusions:
Hybrid stenting is a negotiation between the concept of “full vascular regeneration” and the mechanical limitations of these novel devices. Whether the use of metallic stents, although limited as compared to a full-metal strategy, compromises the benefits of BVS remains however to be discussed. Whether the use of a hybrid strategy with newer (and safer) scaffolds will present any advantage as compared to a full-DES strategy, will need to be studied in the future.
2. They did not mention the clinical implication enough. We would like to know which kind of lesions are appropriate for hybrid strategy or BVS-only strategy. They must analyze and mention them in the result and discussion sections
This is an excellent question, which at the moment can only be partially be answered and only in terms of hypotheses. We added the following text in the discussion:
Beyond any consideration on the safety of the devices implanted, it might be hypothesized that the use of BVS might be more advantageous in long and proximal segments, which might thereafter regain the possibility to adjust their diameter in response to biochemical and physical stimuli. Calcific lesions, in contrast, might have theoretically less potential for regeneration. Thrombotic lesions might also represent a setting for BVS, allowing „plaque stabilization“ as previously reported2. The use of BVS in CTO lesions has also been reported, but no data are available regarding the capacity of these lesions to regenerate3. In the present database, lesions treated with BVS-only were indeed longer and more frequently of type C (thrombotic). There was a trend towards less calcific lesions being treated with BVS-only, but this difference remains speculative. Finally, in theory there is also a rationale for the use of BVS in ostial or bifurcation lesions to limit (in time) the risks associated with malapposed struts, but the evidence on their (lack of) safety in these settings clearly discouraged their use4. Based on the instructions for use, in the present database, bifurcation lesions were almost exclusively treated with metallic stents.
1. Tanaka A, Jabbour RJ, Mitomo S, Latib A and Colombo A. Hybrid Percutaneous Coronary Intervention With Bioresorbable Vascular Scaffolds in Combination With Drug-Eluting Stents or Drug-Coated Balloons for Complex Coronary Lesions. JACC Cardiovascular interventions. 2017;10:539-547.
2. Brugaletta S, Gomez-Lara J, Garcia-Garcia HM, Heo JH, Farooq V, van Geuns RJ, Chevalier B, Windecker S, McClean D, Thuesen L, Whitbourn R, Meredith I, Dorange C, Veldhof S, Rapoza R, Ormiston JA and Serruys PW. Analysis of 1 year virtual histology changes in coronary plaque located behind the struts of the everolimus eluting bioresorbable vascular scaffold. Int J Cardiovasc Imaging. 2012;28:1307-14.
3. Polimeni A, Anadol R, Munzel T, Geyer M, De Rosa S, Indolfi C and Gori T. Bioresorbable vascular scaffolds for percutaneous treatment of chronic total coronary occlusions: a meta-analysis. BMC cardiovascular disorders. 2019;19:59.
4. Gori T, Wiebe J, Capodanno D, Latib A, Lesiak M, Pyxaras SA, Mehilli J, Caramanno G, Di Mario C, Brugaletta S, Weber J, Capranzano P, Sabate M, Mattesini A, Geraci S, Naber CK, Araszkiewicz A, Colombo A, Tamburino C, Nef H and Munzel T. Early and midterm outcomes of bioresorbable vascular scaffolds for ostial coronary lesions: insights from the GHOST-EU registry. EuroIntervention : journal of EuroPCR in collaboration with the Working Group on Interventional Cardiology of the European Society of Cardiology. 2016;12:e550-6.
Round 2
Reviewer 2 Report
Authors have adequately revised the manuscript.
I have no additional comment.
Author Response
Thank you very much for your positive comments.